# Experimental and Computational Studies on Structure and Energetic Properties of Halogen Derivatives of 2-Deoxy-D-Glucose

**DOI:** 10.3390/ijms22073720

**Published:** 2021-04-02

**Authors:** Marcin Ziemniak, Anna Zawadzka-Kazimierczuk, Sylwia Pawlędzio, Maura Malinska, Maja Sołtyka, Damian Trzybiński, Wiktor Koźmiński, Stanisław Skora, Rafał Zieliński, Izabela Fokt, Waldemar Priebe, Krzysztof Woźniak, Beata Pająk

**Affiliations:** 1Biological and Chemical Research Centre, Department of Chemistry, University of Warsaw, Zwirki i Wigury 101, 02-089 Warszawa, Poland; anzaw@chem.uw.edu.pl (A.Z.-K.); spawledzio@chem.uw.edu.pl (S.P.); mmalinska@chem.uw.edu.pl (M.M.); dtrzybinski@cnbc.uw.edu.pl (D.T.); kozmin@chem.uw.edu.pl (W.K.); kwozniak@chem.uw.edu.pl (K.W.); 2Independent Laboratory of Genetics and Molecular Biology, Kaczkowski Military Institute of Hygiene and Epidemiology, Kozielska 4, 01-163 Warsaw, Poland; maja.soltyka@gmail.com; 3Department of Experimental Therapeutics, The University of Texas MD Anderson Cancer Center, 1901 East Rd., Houston, TX 77054, USA; sskora@mdanderson.org (S.S.); RJZielinski@mdanderson.org (R.Z.); ifokt@mdanderson.org (I.F.)

**Keywords:** 2-deoxy-D-glucose, 2-DG analogues, hydrogen bonds, energy frameworks, Hirshfeld surface, NMR spectroscopy, The University at Buffalo Pseudoatom Databank (UBDB), glycolysis inhibitors, electrostatic interactions, ligand binding

## Abstract

The results of structural studies on a series of halogen-substituted derivatives of 2-deoxy-D-glucose (2-DG) are reported. 2-DG is an inhibitor of glycolysis, a metabolic pathway crucial for cancer cell proliferation and viral replication in host cells, and interferes with D-glucose and D-mannose metabolism. Thus, 2-DG and its derivatives are considered as potential anticancer and antiviral drugs. X-ray crystallography shows that a halogen atom present at the C2 position in the pyranose ring does not significantly affect its conformation. However, it has a noticeable effect on the crystal structure. Fluorine derivatives exist as a dense 3D framework isostructural with the parent compound, while Cl- and I-derivatives form layered structures. Analysis of the Hirshfeld surface shows formation of hydrogen bonds involving the halogen, yet no indication for the existence of halogen bonds. Density functional theory (DFT) periodic calculations of cohesive and interaction energies (at the B3LYP level of theory) have supported these findings. NMR studies in the solution show that most of the compounds do not display significant differences in their anomeric equilibria, and that pyranose ring puckering is similar to the crystalline state. For 2-deoxy-2-fluoro-D-glucose (2-FG), electrostatic interaction energies between the ligand and protein for several existing structures of pyranose 2-oxidase were also computed. These interactions mostly involve acidic residues of the protein; single amino-acid substitutions have only a minor impact on binding. These studies provide a better understanding of the structural chemistry of halogen-substituted carbohydrates as well as their intermolecular interactions with proteins determining their distinct biological activity.

## 1. Introduction

2-Deoxy-D-glucose (2-DG) is a synthetic analogue of D-glucose in which a hydroxyl group at C-2 is replaced by a hydrogen atom. Since D-glucose and D-mannose are C-2 epimers and in consequence their deoxygenation at C-2 would produce the same compound, the 2-DG should also be considered an analogue of D-mannose [1] (Figure 1). A significant number of halogen-substituted carbohydrates have been synthesized. Some of them, the fluorine derivatives, were extensively tested, and 2-deoxy-2-fluoro[^18^F]-D-glucose has been used in medicine as an imaging agent for positron emission tomography (PET) [2,3]. Other halogen derivatives have been initially considered as potentially useful intermediates in synthetic carbohydrate chemistry [4], and only later considered and tested as potential biologically active agents [5,6]. A significant scientific effort has recently been allocated to develop new drugs targeting cancer metabolism, particularly glucose metabolism, which differs from glucose processing in normal cells [7,8]. 2-DG is considered a promising anticancer drug targeting glucose metabolism of cancer cells [5,9], including melanoma [10], osteosarcoma [11], and others. Its proposed mechanism of action relies on the inhibition of some key enzymes involved in glucose and mannose metabolism. One of those enzymes is hexokinase (HK), responsible for glucose phosphorylation in the first step of glycolysis. HK inhibition leads to significant impairment of glucose metabolism. The other route to disrupt the homeostasis of cancer cells by 2-DG relies on alteration of protein glycosylation and disruption of signaling pathways [9]. Previous studies demonstrated potential antiviral properties of 2-DG [12,13]. More recent computational studies suggested that 2-DG and its derivatives can block viral proteases, which render the virus unable to bind to its receptor [14]. In summary, so far in addition to 2-DG, only 2-deoxy-2-fluoro-D-glucose (2-FG) has been recognized as a potential anticancer compound in cancer therapy [15,16]. Other halogen derivatives of 2-DG were not extensively investigated. Studies comparing different halogen analogues directly with 2-DG have been initiated by us [6] and these studies and our more recent preliminary data (not published) suggest that other 2-DG derivatives should also be considered as potentially useful agents in the treatment or diagnoses of metabolically dependent diseases.

Despite the potential medical significance of 2-DG derivatives, only a few crystal structures of high quality are available in crystallographic databases (including the parent compound) [17]. This possibly is due to significant difficulties in the preparation of high-quality crystals. Most carbohydrates tend to form amorphic solids, often containing residual solvent. Most of the existing structures are polyfluorinated pyranoses, which are easier to crystallise due to a reduced number of hydroxyl groups [18,19,20]. There is one structure of 2-FG deposited in the CSD database (deposition number 1296336); however, it has only moderate quality. To our best knowledge, no comprehensive structural study investigating the effects of substitution by different halogens at a given position in the pyranose ring has been conducted. A better understanding of the structural features of this class of carbohydrates, both in the solid state and solution, is necessary to achieve an in-depth knowledge of their biological and pharmacological properties. It should be noted that there have been several experimental preclinical and clinical studies with 2-DG (and its derivatives), mainly administrated orally. Thus, the existence of their potential polymorphs may have profound medical implications because different crystal forms usually are characterized by different bioavailability [21,22].

High-resolution structural research combined with theoretical calculations provides a better understanding of intra- and intermolecular interactions in the solid state as well as principal energetic features of the structure, including the interaction anisotropy [23] and charge-density distribution [24]. Such findings may relate to biological and pharmacological properties of drug-like molecules and potential drug candidates [25] as well as crystal engineering [26,27]. The structural scaffolding of both 2-DG and its halogen derivatives is composed of moieties capable of forming a complicated set of hydrogen interactions between hydroxyl groups in the pyranose ring. Moreover, the halogen substitution, particularly by bromine and iodine, creates a possibility for halogen-mediated interactions, which is interesting from the structural perspective because studies on halogen bonding in carbohydrates have been limited due to the lack of available crystal structures. Another interesting possibility is to investigate H-bonding with the fluorine atom in a polyhydroxylated system, which is still a matter of debate in structural chemistry [28].

In this paper, the crystal structures of a series of halogenated 2-DG derivatives (Figure 1), namely, 2-deoxy-2-fluoro-D-glucopyranose (2-FG), 2-deoxy-2-fluoro-D-mannopyranose (2-FM), 2-deoxy-2-chloro-D-glucopyranose (2-CG), 2-deoxy-2-chloro-D-mannopyranose (2-CM), and 2-deoxy-2-iodo-D-glucopyranose (2-IG) have been solved and refined by applying single-crystal X-ray diffraction. We described their crystal properties, focusing on analysis of the H-bond network in the supramolecular landscape. Furthermore, the collected data were used in theoretical calculations to reveal more details concerning their crystal packing as well as intra- and intermolecular interactions. Due to the problems with optimizing the conditions and obtaining the appropriate crystals, 2-deoxy-2-iodo-D-mannopyranose (2-IM), 2-deoxy-2-bromo-D-glucopyranose (2-BG), and 2-deoxy-2-bromo-D-mannopyranose (2-BM) have not been analysed so far. Thus, 2-IM and bromo-derivatives are still under scrutiny in our lab, and the results will be published in a separate paper.

Firstly, we performed Hirshfeld surface analysis to obtain a quantitative and qualitative description of interatomic contacts in the structure. Secondly, we calculated some energetic features of the structure, i.e., the total lattice energy and energy frameworks. Next, high-resolution NMR spectral data shed more light on their structural properties in aqueous solution, including anomeric and conformational equilibria. Finally, we use Hirshfeld analysis and computational methods to investigate the binding of 2-FG in several protein structures quantitatively.

As far as we know, this is the first report describing the structure and energetic properties of halogen derivatives of 2-DG.

## 2. Results and Discussion

### 2.1. Crystallization and Crystal Morphology

Initial attempts to crystallize 2-DG derivatives from water solution resulted only in a syrup-like amorphous phase, thus only crystallization from organic solvents allowed us to obtain a mixture of crystalline and amorphic, vitreous phases. For each compound, the crystalline phase was composed of aggregates in which monomeric single crystals were observed. The crystals displayed an acicular or columnar habit, and most of them were macroscopically twinned.

### 2.2. X-ray Diffraction Studies

Single-crystal X-ray diffraction analysis confirmed the identity of all investigated compounds. Most of them crystallize in the orthorhombic *P*2_1_2_1_2_1_ space group. The only exception is 2-CM, which was forming crystals in the monoclinic *P*2_1_ space group. In all cases, the crystal lattice’s asymmetric unit contained one molecule of compound (Figure 2). The 2-CM turned out to be a solvate. The independent part of its unit cell also included one molecule of water (Figure 2). Details of the crystallographic data and refinement parameters are summarized in Appendix A (ESI). Other parameters such as bond lengths, valence, and torsion angle values can be found in the Electronic Supporting Information (Appendix A).

### 2.3. Conformational Analysis of the Pyranose Ring

A common way to describe a monocyclic ring’s conformation is to use Cremer-Pople parameters (CP), generalized ring-puckering coordinates in a spherical coordinate system. CP parameters allow for the quantitative and precise definition of puckering conformations [29]. In each structure, the carbohydrate molecule forms a pyranose ring existing in the chair ^4^C_1_ (D1) enantiomeric conformer. Calculated CP parameters indicated only minor deviations from the ideal chair geometry (*θ* = 0°), and the largest deviation is observed for 2-CG (*θ* = 7.8°). In all investigated compounds, the geometry of pyranose was also highly similar to the parent compound (*θ* = 6.0° for the *α* anomer), indicating that halogen substitution does not alter the ring conformation in any significant way. The other CP parameters are summarized in Appendix A. The structural similarity of the pyranose rings in 2-DG derivatives was also visualized by structural alignments (Figure 3).

All these compounds crystallise exclusively as *β*-anomers, and the only exception is 2-IG, which forms the *α*-anomer in the crystal state. The C6 hydroxyl group’s orientation is axial in most of the studied compounds except 2-CM, where the C6 hydroxyl group is oriented equatorially. In the crystal, the critical factors driving the conformation of the pyranose ring, as well as the orientation of its substituents, are hydrogen bonds. These non-covalent interactions may stabilize a conformation that is not energetically preferred in an aqueous solution, where the prevalence of a given enantiomeric conformer is determined mostly by the orientation of the largest substituent in the pyranose ring. It is also assumed that both intra- and intermolecular H-bonds between sugar moieties play only a marginal role. 

### 2.4. Intermolecular Interactions

The substitution of the pyranose C2-atom by halogen atoms of various sizes has a noticeable impact on the supramolecular landscape. The significance of changes concerning the mutual arrangement of molecules and the scheme of intermolecular interactions occurring in their crystals is different for individual compounds. However, some general trends can be noticed. In the case of both fluoro derivatives (2-FG and 2-FM), the changes are relatively subtle. A closer look (Figure 4) reveals *β*-2-FG and *β*-2-FM molecules are organized in their crystals almost identically to the *β*-anomer of 2-DG (*β*-2-DG) [30]. Moreover, all three compounds display nearly the same unit cell parameters. This isostructurality results from the relatively small size of the fluorine atom, which is often regarded, when connected to carbon, as an isostere of hydrogen. The existence of such a phenomenon also has been reported for other fluorinated organic systems [31,32]. Differences between the crystal structures of *β*-2-DG, *β*-2-FG, and *β*-2-FM become more apparent after a detailed examination of each structure’s existing scheme of interactions. Sugar molecules are assembled in complex 3D-frameworks, in which a dense network of hydrogen bonds is present.

In the crystal of the parent compound, the adjacent molecules are held together by O−H···O hydrogen bond (d(D···A) = 2.655(2)–2.808(2) Å; <(D−H···A) = 150−172°; Appendix A (ESI)) interactions, and the crystal network is stabilized by the weak intermolecular C−H···O contacts (d(D···A) = 3.380(2)–3.518(2) Å; <(D−H···A) = 146−147°; Appendix A). The fluorine substitution in C2 position results in the intermolecular C−H···F (2-FG) (d(D···A) = 3.325(4) Å; <(D−H···A) = 142°; Appendix A) and O−H···F (2-FM) (d(D···A) = 3.002(4) Å; <(D−H···A) = 142(5)°; Appendix A) interactions, which replace one of the C−H···O contacts observed in the crystal network of *β*-2-DG. Different types of interactions involving halogens in 2-FG and 2-FM stem from different positions of the halogen itself (axial in 2-FM and equatorial in 2-FG). Nevertheless, the overall pattern of the intermolecular O−H···O hydrogen bonds is preserved in all compounds mentioned above, and they differ only by the subtle changes of their geometry. The distances between the donor and acceptor of the H-atoms in hydrogen bonds ranged from 2.622(3) to 2.786(3) Å, and from 2.665(4) to 2.846(5) Å for *β*-2-FG and *β*-2-FM, respectively. Additionally, their <(D−H···A) angles were in the ranges of 149(4)‒175(4)° and 163(5)–171(6)° for *β*-2-FG and *β*-2-FM, respectively (Appendix A).

An increase in the halogen substituent volume leads to the appearance of more evident changes in crystal packing. It is clear that the H-bond pattern in the crystal lattice of all analysed Cl- and I- derivatives is significantly different from the patterns observed in the compounds described previously. Namely, in the case of derivatives bearing either a Cl or I atom, layered entities spreading along the (101) plane can be distinguished (Figure 4). In the crystal structure of *β*-2-CG, molecules are organized in bilayers. Those within this supramolecular assembly are held together by the O−H···O H-bonds (d(D···A) = 2.683(3)−2.771(3) Å; <(D−H···A) = 164(3)−172(3)°)), O−H···Cl (d(D···A) = 3.321(2) Å; <(D−H···A) = 167(2)°) and weak C−H···O H-bonds (d(D···A) = 3.237(3) Å; <(D−H···A) = 128°)) hydrogen bonds (Appendix A).The weak C-H…O interactions involving the adjacent molecules from neighbouring bilayers stabilises the whole crystal structure (d(D···A) = 3.485(3) Å; <(D−H···A) = 157°; Appendix A) of contacts. In the case of the chlorinated mannose derivatives (2-CM), sugar molecules are also arranged in bilayers. However, these layered entities also incorporate water molecules. Within the bilayer, adjacent sugar molecules interact with themselves *via* the O−H···O (d(D···A) = 2.663(3) − 2.815(2) Å; <(D−H···A) = 161(4) − 168(3)°) and C−H···O (d(D···A) = 3.496(3) Å; <(D−H···A) = 157°) hydrogen bonds, and by the O−H···O (d(D···A) = 2.727(2) Å; <(D−H···A) = 175.3(19)°) interactions with the neighbouring water molecules (Appendix A), and these water molecules are also linked by the O−H···O (d(D···A) = 2.629(2) Å; <(D−H···A) = 165(3)°; Appendix A) hydrogen bond. Finally, the 2-CM and water molecules from neighbouring two-component layers are interacting by the intermolecular O−H···O (d(D···A) = 2.851(2) Å; <(D−H···A) = 171(3)°; Appendix A) interactions. It is worth mentioning that in this structure, no hydrogen bonds involving the halogen atoms were identified. This could be associated with the presence of water molecules in the crystal network, which may be better acceptors for H-atoms in this supramolecular framework. 

As mentioned earlier, 2-IG is the only compound among investigated derivatives of 2-DG that crystallised as an *α*-anomer; however, its supramolecular architecture significantly differs from that observed for *α*-2-DG (Figure 4). In the case of the iodo-derivative, the molecules are arranged in distinct layers wherein the O−H···O H-bonds (d(D···A) = 2.648(9)−3.169(9) Å; <(D−H···A) = 127(8)−167(11)°; Appendix A) between adjacent *α*-2-IG molecules were identified. However, the framework of intermolecular interactions in *α*-2-DG is more complex. In this case, adjacent molecules are held together via a dense network of O−H···O (d(D···A) = 2.670(2) − 2.784(2) Å; <(D−H···A) = 155−174°) and C−H···O (d(D···A) = 3.344(3)−3.390(3) Å; <(D−H···A) = 142−148°) hydrogen bonds (Appendix A (ESI)), which leads to the creation of a complex three-dimensional supramolecular assembly [33]. Another aspect of the analysis of the supramolecular landscape was the value of packing indices. Comparison of these values for investigated compounds (69.29-α-2-DG; 68.19%-*β*-2-DG; 70.07-2-FG; 71.27-2-CG; 71.82%-2-IG; 69.77-2-FM and 70.65%-2-CM) revealed the existence of a subtle trend. Here, the increasing value of packing indices correlated with the increasing volume of the substituent. This finding strongly suggests that in the case of halogenated derivatives of 2-DG, the presence of a more bulky substituent leads to a more efficient filling of space in the lattice. 

### 2.5. Hirshfeld Surface Analysis

Hirshfeld surfaces (HS) of molecules in the crystalline state are calculated by partitioning the electron density in the crystal to obtain such a surface where the contribution to the electron density of the crystal from the sum of spherical atoms of a given molecule (promolecule) starts to prevail over the contribution from others molecules in the crystal (procrystal). In other words, for each point belonging to the HS, the contribution to electron density from the promolecule and procrystal is the same. The useful parameters that enable the mapping of interatomic distances are d_i_ and d_e_, which show the closest distance from a point on the HS to an atom nucleus located inside or outside the surface. A convenient way to visualize intermolecular contact in the crystal lattice is to use a two-dimensional fingerprint plot, describing the relation of d_i_ to d_e_ for each point on the Hirshfeld surface, highlighting differences in the crystal lattice among the structures to compare. Furthermore, HS analysis provides a quantitative description of intermolecular interactions in the lattice, allowing the contribution of each type of interatomic contact to the global interaction pattern in the crystal structure. All fingerprint plots are shown in Figure 5, whereas the HS are presented in Appendix A. 

The most distinguishable features of both 2-DG and its halogen derivatives are characteristic, diagonally-symmetric, sharp “spikes,” which confirm the existence of strong O-H···O hydrogen bonds to overall interatomic contacts in the crystal lattice (average estimated sum of d_i_ and d_e_ is from 1.6 to 2.8 Å for all compounds). The location and directionality of the edges are virtually the same for 2-DG and its derivatives, which indicate that halogen substitution does not alter the global pattern of close O···H contacts in the lattice. Furthermore, the substitution is also not significant in terms of contribution of O···H contacts to the HS, which oscillate near 40%. The only minor exception is 2-IG, where the percentage of O···H contacts is 37.5%. For unsubstituted structures, the contribution of H···H contacts is 58.4% and 56.2% for *α*- and *β*-2-DG, respectively. In contrast, the introduction of a halogen atom reduces the percentage of O···H contacts starting from 41.3% (2-FM and 2-CM) to 37.5% (2-IG) in favour of X···H contacts. The lowest contribution is observed for fluorine (13.6%-2-FM and 15.3%-2-FG); for chlorine the contribution is increased (18%-2-CM and 19.6%-2-CG), achieving the highest percentage of 20.6% for 2-IG. This observation is mostly related to the increasing size of the halogen substituent. Moreover, fingerprint plots of 2-CG and 2-CG show a high number of longer (d_i_ + d_e_ > 3.5 Å) O···H and H···H contacts, which stipulate slightly dense crystal packing, and this tendency is even more pronounced in the case of 2-IG. The contribution of other contacts, namely, O···O, O···X, and X···X, is mostly marginal. These contacts are rather long-range and do not play an important role in the lattice. A somewhat surprising finding is the almost total lack of any halogen-halogen only contacts in the investigated structures except for minor I···I interaction in 2-IG. Still, their contribution is minimal (below 1.5%). 

The shapes of fingerprint plots for 2-DG and its fluorine derivatives bear a strong resemblance, mirroring their isostructural crystal lattices. The only discernible difference among those compounds is the presence of scattered, medium-range (d_i_ + d_e_ > 2.3 Å) H···H contacts in 2-FM. Otherwise, the patterns of H···H contacts overlap each other in both F-derivatives and do not differentiate from the H···H contact pattern in both anomers of 2-DG. These observations are in line with the literature showing that, in general, fluorine is considered to be a good mimic of hydrogen in the crystalline state. Both 2-FG and 2-FM form several medium-range H···F contacts scattered in the center of the fingerprint and overlapping with H···H contacts. The presence of secondary “spikes” of H···Cl contacts in the fingerprint plot for 2-CG indicates the presence of relatively strong O-H···Cl bonding between sugar molecules, and the edges of the spikes are around d_H···Cl_ ≈ 2.3 Å. This feature is less pronounced in the case of 2-CM, where H···Cl interactions are significantly longer and the shortest H···Cl contacts are around 2.8 Å. The shape of the fingerprint for a water molecule in 2-CM confirms strong O-H···O bonding, and only a minimal contribution of H···Cl contacts. 

### 2.6. Cohesive Energies and Energy Frameworks

Total cohesive energy for each compound has been calculated using the *Crystal09* software package (Table 1).

In the series of Glc derivatives, the energy slightly increases with the increasing size of the substituents, which shows that a steric hindrance introduced by a halogen atom leads to slight destabilization of the lattice. Nevertheless, this effect is not very noticeable. The total energy for 2-DG and 2-FG (*β*-anomers) is almost the same and this finding is consistent with their isostructurality. Interestingly, our calculations showed that the cohesive energy of *α*-2-DG is slightly higher than its second anomer, which indicates lower stability of this form and the observable tendency for this class of pyranoses to crystallize as *β* anomers. The only exception is 2-IG, where a bulky substituent forces the preference toward an *α* anomer. It is difficult to discuss the differences in stability between anomers of studied compounds owing to the limited data that are available. However, both *α* anomers have higher energies than other compounds that crystallize as *β* anomers, which may indirectly suggest generally higher stability of *β* anomer in the crystal state. Interaction energies between molecules in the lattice and overall energy frameworks have been calculated using *Crystal Explorer* (Figure 6). In any case, given sugar molecules are involved in several different interactions within a cluster of adjacent molecules. Most of the interaction is mediated by H-bonds, involving halogen atoms (either F or Cl). The isostructurality of *β*-2-DG and its F-derivatives is manifested in the dimer interaction energies extracted from *Crystal Explorer* calculations (see Appendix A). Most energetically important dimers are created (symmetry relation −*x* + 1, *y* − ½, −*z* + ½) via O5···HA(O3) and O1···H4(O4) H-bonding, and another significant contribution to lattice energy is from O6···H1(O1) and O3···H6(O6) bonds (symmetry relations are *x* − 1, *y*, z and −*x* + 3/2, −*y* + 1, and *z* + ½, respectively). 

The directionality and general pattern of these energy frameworks were very similar (Figure 6). Interestingly, additional stability of 2-FM is likely to be caused by the F1···H6(O6) bond, which is not present in 2-FG. In the latter structure, H-F interactions are limited to a relatively weak F1···H6(C6) bond, which is also visible on corresponding Hirshfeld surfaces and fingerprint plots (Appendix A). The energy framework calculated for *α*-2-DG is also dense. However, it displays different directionality, which is directly caused by different H-bonding geometry in the lattice. The most important contribution is from H-bonds O6···H4(O4) and O5···H1(O1).

The lattice of 2-CG is mostly stabilized by H-bonding involving the C1, C3, and C6 hydroxyl groups, which forms a relatively dense network of interactions leading to the formation of a bilayered supramolecular entity. A somewhat exotic Cl···H6(O6) bond provides only a minor contribution to lattice stabilization. The bilayers are held together mainly by weak dispersive C···H interactions. Due to the presence of water, the energy framework for 2-CM is denser and, to some extent, bears a resemblance to frameworks calculated for 2-FG and 2-FM. As in the case of most hydrates, water-sugar interactions contribute significantly to lattice stabilization. H-bonds O5···H3(O3) and O4···H6(O6) along the a-axis provide the most crucial contribution to stabilization energy. Bilayers interact with one another via O3···H4(O4) and O6···H4(C4) bonds, which are less energetically significant and the latter of which is more dispersive if compared to other H-bonds in the lattice. Owing to substantial steric hindrance related to the iodine atom, the energy framework for the 2-IG lattice is predominantly excluded from interactions within 2-dimensional monolayers. The most significant contributions are from x ± 1, y, z, dimers (O5···H3(O3) and O3···H6(O6) bonds) and, to a lesser extent, x, y ± 1, z dimers, which form other H-bonds. Interactions between the monolayers are much weaker and are mediated by H···H and H···I contacts, and a significant energetic contribution is from the dispersive term. In addition to the mentioned studies, we also calculated the enrichment ratios (E_XY_) for interatomic contacts whenever they were statistically significant. In all compounds the E_OH_ was higher than 1.0, which is caused by the formation of energetically favourable H-bonds. A similar trend was evident for hydrogen-halogen contacts where E_XH_ was also higher than 1, indicating that halogen is preferred in contacts with H atoms, which is preferred from an electrostatic (F, Cl) or dispersive (I) point of view (see Tab S26 for details). These findings further strengthen previously mentioned observations from general analysis of HS and energy frameworks.

### 2.7. Structure in Aqueous Solution

Apart from the compounds mentioned before in the study, we also included 2-deoxy-2,2-difluoro-D-glucopyranose (2,2-diFG) and 2-deoxy-2-iodo-D-mannopyranose (2-IM) to gain better insight into how the halogen substitution at C2 may alter the conformation of the pyranose ring as well as anomeric and rotamer equilibria. 

Due to the endo-anomeric effect, caused mainly by electrostatic dipole-dipole interactions between the O1 and O5 oxygens, the axial (*α*) anomer’s contribution is higher than would be expected from simple conformational analysis [34,35]. For the non-modified glucose and mannose in their pyranose forms, the contribution of the *α*-anomer is 37% and 69%, respectively [36]. It is known that preference for axial orientation of the C1 substituent in D-glucopyranose tends to increase along with increased electronegativity of the mentioned substituent. Nevertheless, the nature of a substituent and configuration in the C2 position also influences the magnitude of the anomeric effect [37]. In the case of our glucose derivatives, the impact of halogen substitution in the C2 position is visible only for fluorine-substituted compounds; this is due to the very high electronegativity of fluorine (see Appendix A). 

Due to the presence of a CF_2_ moiety at the C2 position, this effect is more significant for 2,2′-FG than for 2-FG, and the contribution of the α-anomer is 43% and 68%, respectively. In this regard, 2,2′-diFG behaves as an mannose derivative. The prevalence of an energetically unfavoured α-anomer in mannopyranose results from bisection of the torsional angle between the C1-O1 and C1-O5 bonds by the 2’OH group. This so-called Δ2 effect introduces an additional dipolar interaction, which increases the overall electronic destabilization in the pyranose ring, leading to a higher anomeric effect [38]. Because 2-DG is devoid of that OH group, the steric torsion is alleviated, and the α-anomer contribution is reduced to 49%. In contrast to the mentioned glucose derivatives, electronegativity seems to have little influence on anomeric equilibria in mannose derivatives. A more important factor is the size of a substituent. More bulky halogen atoms cause significant steric torsion, which competes with the dipolar effects shifting the equilibrium toward the β-anomer [39]. 

In an aqueous solution, pyranosides exist as a mixture of ^1^C_4_ and ^4^C_1_ conformers. However, in most situations, the equilibrium is strongly shifted toward the more energetically stable ^4^C_1_ isomer. Several other, less energetically stable conformations may also exist in solution, and the conformational equilibria may be quite intricate [34]. To gain better insight into the studied compound’s behaviour in an aqueous solution, we employed NMR spectroscopy to collect both 1D ^1^H and ^19^F spectra and ^13^C HSQC and NOESY correlation spectra. 

^3^*J*_H,H_ constants in the studied compound were derived from either HSQC or ^1^H spectra (see Appendix A for additional data on ^19^F spectra), which allowed us to determine most of the H-C_X_-C_Y_-H torsion angles in the pyranose ring using the expanded Altona equation, which takes into account electronegativity of the neighbouring substituent (see Material and Methods for the details) [40]. The results summarized in Appendix A indicate that all studied compounds exist mostly in the strongly prevalent ^4^C_1_ conformation. These findings were supported by H-H distances calculated from the NOESY spectra. Due to the somewhat limited accuracy of a simple conformational analysis based on exclusively ^3^*J*_H,H_ constants and ^1^H NOE data, we were unable to estimate the contributions of other possible conformers (i.e., ^1^C_4_ or twisted chair) or acyclic form. The differences in the pyranose ring geometry between 2-DG and its derivatives are minimal, mostly limited to H1-C1-C2-H2 and H2-C2-C3-H3 torsion angles, which are directly influenced by the substituent. These deviations slightly correlate with the halogen atom size but do not alter the overall geometry of the ring. Conformations in crystal state and in solution also display high similarity. The observed differences are mostly caused by intermolecular H-bonds and dispersive interactions in the crystalline state, which play only a marginal role in the aqueous solution. Observed differences in the H-H distances between solution and crystal structures are caused mostly by two factors. Firstly, the structure in solution is more dynamic. As various conformations are adapted in a fast-exchange regime, for both NOESY factors and scalar coupling values the population-averaged values are observed, hence only an average picture of the conformation is available. Due to the CH_2_OH group’s free rotation in solution, it is especially pronounced for the distances between H4-H5 and H5-H6 protons whose positions are fixed for the crystal structures. Secondly, the C-H bond length in the crystal structures solved by X-ray is, in most cases, shorter than determined by high-precision spectroscopic methods or neutron diffraction (0.97 Å *vs* 1.09 Å on average). The difference is caused by the independent atom model (IAM) limitation in the structure refinement in X-ray crystallography [41]. Even if this difference is considered in the data normalization in the NOE analysis, the moderate precision in determining H-X bond in IAM refinement may hinder the conformational analysis.

To estimate the relative distribution of the *gt*, *gg*, and *tg* C5-C6 rotamers, we applied formalism proposed by Stenutz et al. derived from analysis of ^1^*J*_CH_, ^2^*J*_HH_, and ^3^*J*_HH_ coupling constants using limiting values for ^3^*J*_H5, H6R_, ^3^*J*_H5, H6R_, and ^2^*J*_H6R, H6S_ derived from a set of general Karplus equations (see Materials and Methods for details) [42]. Their position within the pyranose ring halogen substitution has no measured effect on the rotamer population compared to the parent compound. On the other hand, a slightly higher contribution of the *gt* is observed for the *β* anomer for each of the halogenated analogues and 2-DG. Nonetheless, these apparent differences approach the accuracy of the conformational analysis. Without more detailed studies beyond the scope of this work, it is unlikely to provide a definitive explanation of this phenomenon. We did not observe any structural disorder in the crystal structure that may indicate any other rotamers than reported in Appendix A. The existence of a given rotamer is imposed merely by the crystal packing and a network of H-bonding in the crystal lattice. It may be different from the rotamer equilibria in solution. 

### 2.8. Protein-Ligand Interaction

Since there are several protein-ligand (15 January 2021) complexes containing 2-FG in the PDB database, we were interested in unravelling some details involving the nature of protein-ligand interactions in these structures, including mapping of interatomic contacts as well as electrostatic interaction energies (*E*_el_). We selected a few exemplary structures using the following criteria: conformation similar to that existing in the solid and aqueous states, lack of covalent bonds with the macromolecule or other ligands, and high quality of the structure, especially for the catalytic centre. We decided to focus on point mutants of fungal pyranose 2-oxidase (P2Ox) participating in lignin degradation by producing H_2_O_2_, which is needed for peroxidases involved in lignin metabolism [43]. P2Ox oxidizes aldopyranoses derived from cellulose and hemicellulose at the C-2 or C-3 position to the corresponding ketoaldolases [44]. The 2-FG is a slow substrate for this enzyme; hence it found some application in its biochemical and structural characterization [45,46,47]. Since the wild-type structure with 2-FG is not available, we decided to use F454N mutants because this residue is not directly involved in substrate binding. In the F454N mutant, the 454N residue does not interact directly with the substrate and thus does not contribute to binding energy, whereas 454F in the other structure displays the same behaviour. Nevertheless, the substitution leads to the substrate loop’s increased flexibility, which may indirectly affect the binding [46]. 

The catalytic centre structure and most notable protein-ligand interactions are depicted in Figure 7A. Structural alignments of all protein structures showed an overall high degree of similarity between them regarding the catalytic centre (Appendix A), which was mentioned previously [46,47]. In all investigated complexes, 2-FG exists in the ^4^C_1_ conformation, and the geometry of the pyranose ring is very similar in each structure. Moreover, these geometries also show similarity to the structure present in the crystalline state, except for different anomeric forms (Figure 7B). 

To achieve an in-depth understanding of interaction in macromolecular structures, charge-density distribution, which may be computed using several quantum-mechanical approaches for small molecules, was taken into account. Then, charge density distribution can be transferred to bigger molecules, e.g., proteins. These methods assume that atoms located in similar chemical environments have similar charge density distributions, which can be emulated by a set of aspherical pseudoatoms. We decided to use a University at Buffalo Pseudoatom Databank (UBDB) composed of theoretically derived pseudoatoms assigned to corresponding atoms in the ligand and protein. It allowed us to obtain *E*_el_. using the exact potential and multipole model (EPMM). We then calculated ligand-interaction energies for each amino-acid residue and then focused on selected residues located near the ligand molecule (Table 2). 

The aims were (1) to find which residues contribute the most to binding and compare our finding with qualitative analysis in the literature, and (2) to check whether these interaction energies are relevant to some macroscale kinetic or thermodynamic parameters such as Michaelis constants (*K*_m_), which were determined previously (Table 3).

The most significant contribution to binding energy comes from R472, which is not involved in direct H-bonding However, it shows that the negatively charged FDA molecule influences charge density distribution in the ligand with a formal neutral charge. Therefore, low *E*el for this residue is a consequence of charge redistribution from FDA to 2-FG. It also explains the finding of the repulsive character of D452 interactions, which form an H-bond with the ligand. In this case, it may be explained by electrostatic and steric factors since the localization of the residue in the binding cavity impedes the ligand’s preferable geometry. A similar situation is observed for the FDA, where flavin interacts with the 4OH group and H atoms in the pyranose ring. The contribution of other amino-acid residues is less significant. It comes from strong H-bonds (Q448, V546, H548, and N593) or medium- to long-range interactions of a dispersive character as H···H (F474, L361) or C-H···O (T169) contacts. The destabilizing effect of H450 and L545 is predominantly caused by H···H and H···O repulsive interactions, respectively. Comparison between interaction energies among P2Ox mutants shows that only in double mutants the energy is significantly lower, and there is no evident correlation between interaction energy and kinetic parameters. However, it needs to be mentioned that *K*_m_ and *k*_cat_ are kinetic parameters showing the concentration of the substrate, in which half of the maximal reaction velocity and the maximum conversion of the substrate molecules per second (for a single catalytic site) can be achieved. On the other hand, the dissociation constant (*K*_d_) is a thermodynamic parameter directly showing the affinity of a ligand to its molecular target and the relationship between the parameters mentioned above. Unfortunately, values of *K*_d_ for 2-FG to P2Ox are not available in the literature, which obfuscates a more detailed analysis of the relationship between *E*_el_ and ligand-protein interactions in this case. 

Using a different approach to study ligand-protein interaction for all structures, we generated HS and fingerprint plots for 2-FG bound in the catalytic centre (Figure 7, Appendix A). We included the same residues that were discussed in the previous paragraph (Table 3). The distributions of interatomic contacts are the same for each structure (Appendix A), showing that one or two single substitutions have negligible effects on the interaction patterns, hence the following discussion concerns all studied complexes. A fingerprint plot for the 3K4L structure is depicted in Figure 6C; the pattern of contacts is similar when compared to the crystalline state (see Appendix A). The main difference is a high number of long-range interactions in the protein complex caused by the less dense packing and the removal of water molecules before interaction energy calculations. Otherwise, the shapes of the fingerprint are very alike in both instances because the most important contributions come from H-bonding and H···H short contacts. The nature of H-bonds in the complex (rendered as a dotted line in Figure 6B) stemmed from different chemical compositions of the adjacent residues (see Appendix A for details). In three H-bonds, the donors are within the 2-FG molecule (O1, O3, and O4) and interact with the COO- group of D452, C=O group of V546, and nitrogen atom from the histidine ring of H548, respectively. The O3 atom in FG-2 is an acceptor of an H-atom belonging to the NH_2_ group of N593. Interestingly, a fluorine atom is involved in H-bonding, acting as an acceptor of an H atom from the NH2 group of Q448. This H-bond was not recognized in the previous literature, and its existence explains the relatively high contribution of Q448 to the total interaction energy. Due to the presence of strong N-H hydrogen bonds, the percentage of N···X contacts between 2-FG and its surroundings is relatively high, and the contribution of O···H contacts is reduced if compared to the solid-state crystalline structure. Another difference is a higher contribution of H···H contacts in the complex, which is mostly caused by the presence of H-rich aliphatic fragments and aromatic rings in the vicinity of the ligand.

## 3. Materials and Methods

### 3.1. X-ray Data Collection and Structure Refinement

Good-quality single-crystals of 2-FG, 2-CG, 2-IG, 2-FM, and 2-CM were selected for the X-ray diffraction experiments at *T* = 100(2) K. Diffraction data were collected on the Agilent Technologies SuperNova Dual Source diffractometer with Cu*Kα* radiation (*λ* = 1.54184 Å) using CrysAlis RED software (CrysAlisPRO, Oxford Diffraction/Agilent Technologies UK Ltd., Yarnton, England). In all cases, the analytical numerical absorption correction using a multifaceted crystal model implemented in SCALE3 ABSPACK scaling algorithm, was applied [48]. The structural determination procedure was carried out using the SHELX package [49]. The structures were solved with direct methods and then successive least-square refinement was carried out based on the full-matrix least-squares method on *F*^2^ using the SHELXL program. All hydrogen atoms linked to oxygen atoms were located from the Fourier difference electron density maps and refined with *U*_iso_(H) = 1.5O_eq_(O). The DFIX 0.82 and 0.85 restraints were applied respectively for the O−H distances of hydroxyl groups of sugar and water molecules. The distance between adjacent H-atoms within the water molecule in 2-CM was restrained to 1.39 Å. All remaining H-atoms were positioned geometrically, with C–H equal to 0.97 and 0.98 Å for the methylene and methine H-atoms, respectively, with *U*_iso_(H) = 1.2 *U*_eq_(C). All molecular interactions in the crystals of the investigated compounds were identified using the PLATON program [50]. The figures for this publication, regarding the structures in the crystalline state, were prepared using the Olex2 and Mercury programs [51,52]. 

### 3.2. Hirshfeld Analysis and Crystal Lattice Energy

The optimizations of the molecular geometries at fixed lattice parameters of -F, -Cl, and -I derivatives were performed using *Crystal09* with the DFT method and B3LYP functional [53]. We applied 6–31 G(d,p) basis set [54] for H, C, N, and O and pob-TZVP basis set [55,56] for F, Cl, and I atoms. The crystal lattice energies were calculated with Grimme dispersion [57,58] and BSSE corrections available in the used version of the software and according to the manual provided by the software developer (Crystal Solutions) [59]. Energy frameworks were calculated with the B3LYP functional and 6–31 G(d,p) basis set using CrystalExplorer17.5 (University of Western Australia, 2017) [23]. Hirshfeld surfaces [60] and two-dimensional fingerprint plots [61] were also calculated using CrystalExplorer17.5 and the atomic coordinates used in the calculations were taken from the given crystallographic data set. The enrichment ratios of interatomic contacts were calculated accordingly to the literature [62] using the following equations:(1)$$S_{X}=C_{XX\;}+\frac{1}{2}\;\underset{Y\neq X}{\sum }C_{XY}$$
(2)$$R_{XY}=2S_{X}S_{Y}$$
(3)$$E_{XY}=\;C_{XY\;}/R_{XY}$$
where *C**_XY_* is a contribution of a given type of atomic contact to the Hirschfeld surface and *E**_XY_* is the enrichment ratio.

### 3.3. Electrostatic Calculations (Ligand-Protein Complexes)

Pseudoatom databanks allowed for the reconstruction of the electron density of macromolecular systems for which experimentally derived geometries are available. In this study, we used UBDB [63] and LSDB to transfer the multiple parameters of the atom types stored in UBDB for the protein-2-FG complexes. 

#### 3.3.1. Preparation of the Protein Structures

Molecule 2-FG as a ligand was found in 35 PDB entries [64]. We selected only crystal structures with measured resolution better than 2.5 Å, and G2F as a standalone ligand. After applying selection criteria, pyranose-2-oxidase H450G mutant (pdbid: 4mof), F454N mutant (pdbid: 3k4l), and H450G/V546C double mutant (pdbid: 4moj) were considered for further studies. We first used Reduce software to add hydrogen atoms to water molecules, protein residues, and ligands to optimize the hydrogen bond network for all the analysed PDB structures. Then, water molecules were removed. Arg, Lys, Asp, and Glu residues were treated as ionized. All amino-acid residues and molecules of 2-FG were scaled independently to their formal charges after the databank transfer. 

#### 3.3.2. Electrostatic Interaction Energy between Ligand and Protein

To obtain the electrostatic interaction energy (*E*_el_) between drug and receptor, the exact potential and multipole model (EPMM) [65] was applied, which allowed computation of *E*_el_ between two molecular charge distributions represented within the Hansen–Coppens electron-density formalism [66]. It combines a numerical evaluation of the exact Coulomb integral for short-range interatomic interactions (less than 4.5 Å) with a Buckingham type multipole approximation for the long-range contacts. After generating charge density distributions of selected complexes with the aid of the UBDB, the EPMM method was executed in XDPROP [67]. The chains were analysed separately. 

#### 3.3.3. Geometry of Ligand Binding and Molecular Visualisation

HS and fingerprint plots for the ligands coordinated by amino-acid residues were calculated in the same manner as for the crystal structures described earlier, starting with the same prepared PDB structures used for electrostatic calculations. All further manipulation of the structures (structural alignments, finding H-bonds, superimposed ligand geometries), as well as images of protein-ligand interactions, were created using UCSF ChimeraX [68].

### 3.4. NMR Spectroscopy

#### 3.4.1. Data Collection and Processing

All NMR experiments were conducted on a 600 MHz Agilent DD2 spectrometer equipped with a standard 5 mm HCN triple-resonance room temperature probe head. Measurements were performed at a temperature of 298 K. All 1D experiments were displayed and analysed in the MNova (Mestrelab Research) program, and 2D in MNova or UCSF Sparky [69]. For resonance assignment, a set of three experiments for each sample was used (i.e., COSY, ^13^C-HSQC, and HMBC). For the measurement of scalar coupling constants, two types of experiments were used: ^1^H 1D spectrum (with water suppression) for well-separated resonances (peaks not overlapping with others) and a modified version of ^13^C-HSQC (below, called ^13^C-HSQC-AP) for signals overlapped in a 1D spectrum. The ^13^C-HSQC-AP was performed with no carbon decoupling during the acquisition period, which allowed for a long acquisition time (of 1.2 s), leading to narrow spectral linewidths, crucial for *J*-coupling determination. The coherence at the beginning of the acquisition time was anti-phase proton magnetization (with respect to carbon). During acquisition time, it was gradually converted into an observable in-phase proton magnetization. Such a scheme allowed us to obtain peaks of a proper phase, independently of the proton-carbon scalar coupling constant used for coherence transfer. Determination of *J*-coupling constants using 1D data sets was performed by line fitting, which allowed us to accurately determine small coupling constants in a case of partially overlapping multiplet components (often observed for pairs of protons at C1 and C2). Determination of *J*-coupling constants using 2D spectra was performed using peak positions, without using any fitting procedure. NOE factors were determined from a set of 2D NOESY experiments for each compound. The set consisted of spectra recorded for six different NOESY mixing time values, i.e., 100, 200, 350, 500, 600, and 750 ms. For each spectrum, cross-relaxation rate constants *σ* (equal to the slopes of the dependencies of a given correlation peak intensity vs. mixing time) were determined. The distances between nuclei (*r*) were determined based on a reference distance *r_ref_* between the two protons of the methylene group at the C6 atom in the *β* form of each sugar, assumed to be equal to 1.77 Å (on the base of the geometrical structure of the pyranose ring), using the following formula:(4)$$r=r_{ref}{\left(\frac{\sigma }{\sigma_{ref}}\right)}^{-1/6}$$
where *σ**_ref_* is the cross-relaxation rate of a reference signal. Relevant experimental parameters for all experiments were gathered in Table 4. 

#### 3.4.2. Conformational Analysis

H-C_X_-C_Y_-H torsion angles in the pyranose ring in the solution were calculated using the following equation:(5)$${\;}^{3}J_{H,H}=C_{0}+C_{1}cos\left(\phi \right)+C_{2}cos\left(2\phi \right)+C_{3}cos\left(3\phi \right)+S_{1}sin\left(\phi \right)+S_{2}sin\left(2\phi \right)$$
where φ is a torsion angle and coefficients *C_i_* and *S_i_* are either numerical constants or functions of relative electronegativities *λ_i_* of substituents in C_X_ and C_Y_ atoms themselves. Values of the above-mentioned parameters were taken from the literature [40,70,71], where the detailed derivation of the aforementioned equation and its components is also provided. 

C5-C6 rotamer populations were estimated using the following set of linear equations:(6)$${\;}^{3}J_{H5,H6R}=9.9A+0.8B+4.5C$$
(7)$${\;}^{3}J_{H5,H6S}=1.5A+1.3B+10.8C^{\;}$$
(8)$$A+B+C=1^{\;}$$
where number coefficients are limiting values of ^3^*J*_*H*5,*H*6*R*_, ^3^*J*_*H*5,*H*6*R*_ coupling constants for a given rotamer and parameters *A*, *B*, and *C* are molar fractions of *gt*, *gg,* and *tg* rotamers, respectively. 

C6-O6 rotamer populations were estimated using a following set of linear equations:(9)$${\;}^{2}J_{H6R,H6S\left(gauche\right)}=12.7\left(A+B\right)+11.5C$$
(10)$${\;}^{2}J_{H6R,H6S\left(trans\right)}=9.7\left(A+B\right)-8.5C^{\;}$$
(11)$${\;}^{2}J_{H6R,H6S\left(app\right)}={\;}^{3}J_{H6R,H6S\left(gauche\right)}D+{\;}^{3}J_{H6R,H6S\left(trans\right)}E^{\;}$$
(12)$$D+E=1^{\;}$$
where number coefficients are limiting values of ^2^*J*_*H*6*R*,*H*6*S*(*gauche*)_, ^2^*J*_*H*6,*H*6*S*(*trans*)_ coupling constants for a given rotamer, *A*, *B*, and *C* are molar fractions of *gt*, *gg*, and *tg* rotamers, respectively, and *D* and *E* are molar fractions of *gauche* and *trans* rotamers, respectively. Values of all torsion angles and H-H distances for the crystal structures were calculated in UCSF Chimera [72]. 

## 4. Conclusions

Our structural research showed that halogen substitution at the C2 position has a significant effect on the crystal structure of the pyranoses; however, the structure of the ring itself is mostly unaffected by the presence of the halogen atom, which was quantitatively shown using CP parameters. The changes in the crystal structure are mostly caused by steric effects because increasing size of the substituent leads to formation of a layered structure having higher packing indices. In spite of more efficient space filling, the H-bond network is less dense in chloro- and iodine derivatives due to higher volume of the substituent itself. Since fluorine is considered to be a good mimic of both hydroxyl groups and hydrogen in many systems, crystals of both F-derivatives have very similar structure to the *β* anomer of 2-DG. Neither packing nor pattern of H-bond is altered in any significant way. To achieve more profound effects on crystal structure, it is necessary to introduce more F atoms to the pyranose ring, which has been extensively studied in earlier studies. In some instances, the F or Cl substituent is capable of forming H-bonds as an acceptor, which suggests (in the case of 2-FG and 2-FM) that fluorine is more isosteric with OH groups than with hydrogen atom. We found that Hirshfeld partitioning is a useful tool in the quantitative analysis of interatomic interactions in systems having intricate networks of H-bonds and other short- and medium-range interactions. Our analysis indicates that the general pattern of interatomic interactions is preserved upon halogen substitution; however, some of the H···H contacts in the parent compound are replaced by H···X (where X is a halogen) contacts. The only noticeable alteration is an increasing number of medium- to long-range interactions between H and X, which is positively correlated with increasing volume of X. None of the studied compounds possesses halogen bonds within their crystal structures, even if the halogen atoms are located in the relative vicinity, as in the case of 2-IG. It can be noticed that the major contributions to intermolecular interaction in the structure are attributable to H-bonds and short-range H···H(X) contacts whilst other interactions (i.e., O···X or X···X) are marginal. These observations are further supported by the energy framework calculations showing that strong H-bonds between H and O provide the most substantial contribution to the lattice energy, and the increasing contribution of dispersive term is positively correlated with the size and polarizability of the substituent. The directionality of the frameworks also confirms the layered structure of Cl and I derivatives, where the mono- or bilayers are composed of strongly interacting molecules but interactions between the layers themselves are relatively weak, mostly dispersive of the H···H or H···X type. Despite the different chemical nature of substituents, the main factor contributing to changes in the crystal structure is the size of the halogen atom, which imposes the formation of a layered supramolecular landscape. 

NMR spectra showed similarity between pyranose puckering between 2-DG and the studied derivatives. Introduction of a halogen atom does not impose any significant conformational changes, and all compounds exist in the most stable ^4^C_1_ chair conformation. As only averaged NMR parameters are observable, it is difficult to estimate the contribution of minor conformation states or chain structures. However, comparison of the NMR and X-ray data indicates that the structure observed in crystal is predominant also for the respective anomer in solution. To achieve higher precision in the calculation of torsion angles, we applied an extended Karplus-like equation, which takes into account the effects of electronegativity of adjacent substituents. The differences in torsion angles are not significant and stemmed from different chemical environments between solvated molecules in solution and molecules in the crystal; in the latter, the directionality of H-bonds and dispersive interactions play significant roles whereas their roles are negligible in solution. The direct comparison of intramolecular H-H distances between molecules in the crystal state and solution is difficult due to difference in C-O and C-H bonds inferred from spectroscopic and X-ray crystallographic experiments. Nonetheless, our NOESY data also indicate that the ^4^C_1_ conformation prevails in aqueous solution. The only significant structural difference among studied compounds (in aqueous solution) was the *α*/*β* anomer ratio, which is caused by stereoelectronic effects invoked by halogen substitution. 

Using a few chosen PDB structures of P2Ox, we were able to extract interaction electrostatic energies between the 2-FG and protein residues within the binding cavity and thus obtain the overall interaction energy between the ligand and protein. However, due to the lack of data on any thermodynamic parameters (e.g., *K*_i_ or Δ*H*), it was difficult to relate these findings to ligand binding because the calculated energies were similar to one another. Analysis of the Hirshfeld surface showed that the chemical environment of 2-FG in the binding cavity is similar to the one in its crystal structure; we also found that a H-F hydrogen bond plays an important role in the binding, a detail omitted by previous studies on that system. 

## Figures and Tables

**Figure 1 ijms-22-03720-f001:**
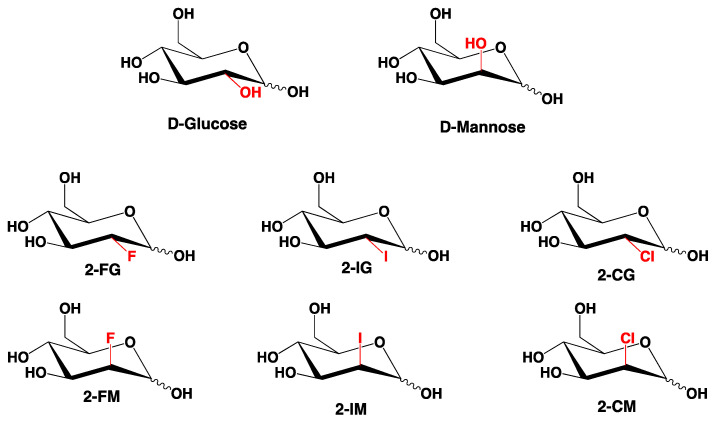
Chemical structures of investigated compounds. These compounds can be considered as 2-substituted D-glucose (Glc) or D-mannose (Man) derivatives.

**Figure 2 ijms-22-03720-f002:**
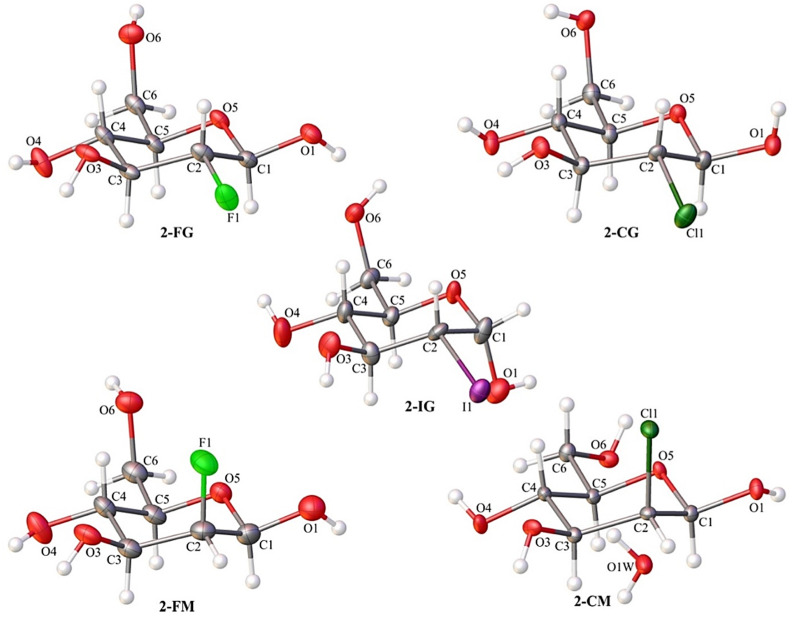
The content of the asymmetric unit of the crystal structures of investigated compounds with the atom numbering scheme. Atomic displacement ellipsoids are shown with 50% probability and the H-atoms are shown as small spheres of arbitrary radius.

**Figure 3 ijms-22-03720-f003:**
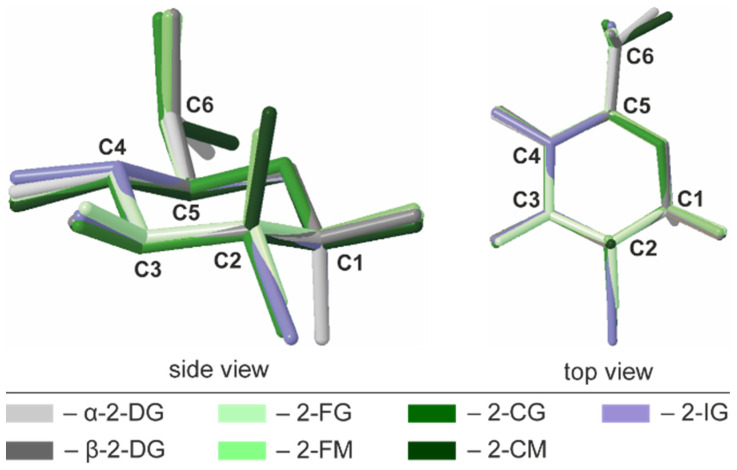
Superimposed structures of investigated halogen derivatives compared to *α*- and *β*-anomers of their parent compound.

**Figure 4 ijms-22-03720-f004:**
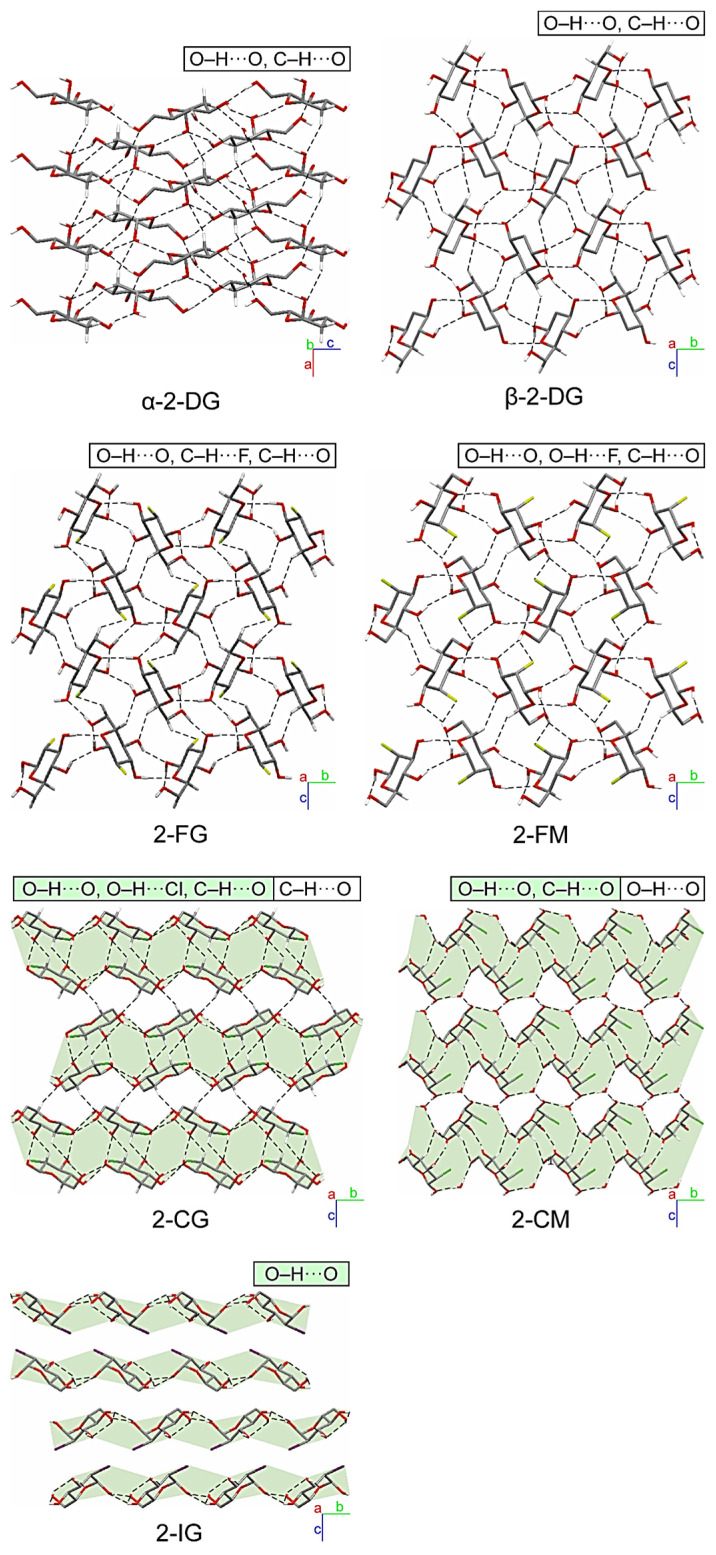
Comparison of packing of molecules in the crystals of investigated halogenated derivatives of glucose and mannose, and the *α*- and *β*-anomers of their parent compound. Hydrogen bonds are represented by the dashed lines. The H-atoms not participating in intermolecular hydrogen bonds have been omitted for the sake of clarity. Layered entities have been highlighted in light-green.

**Figure 5 ijms-22-03720-f005:**
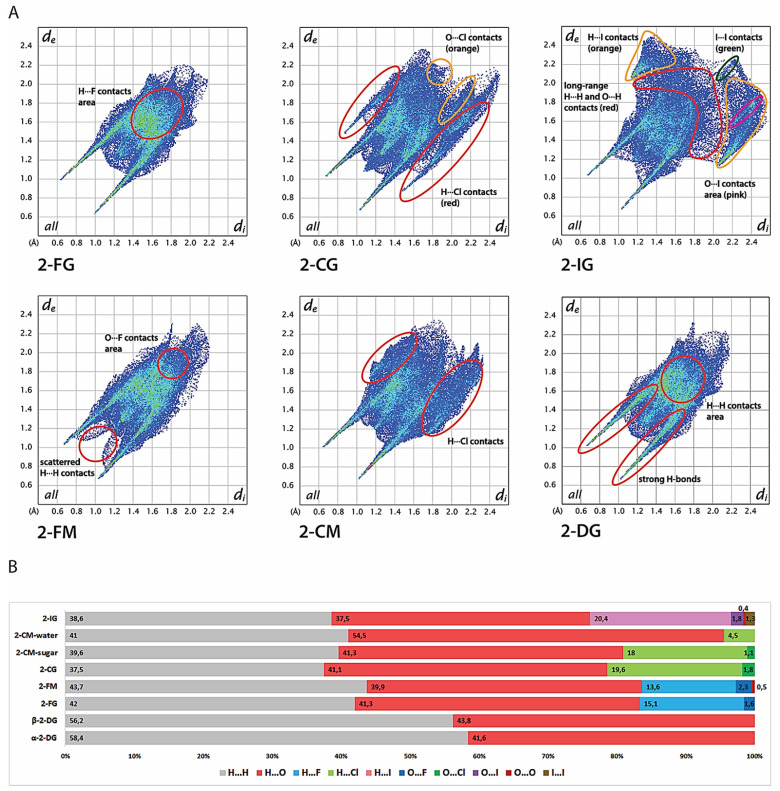
Hirshfeld analysis of the crystal structures. (**A**) Two-dimensional fingerprint plots for the analysed crystals. (**B**) Contribution of each type of interatom contact to the overall Hirshfeld surface for each crystal lattice.

**Figure 6 ijms-22-03720-f006:**
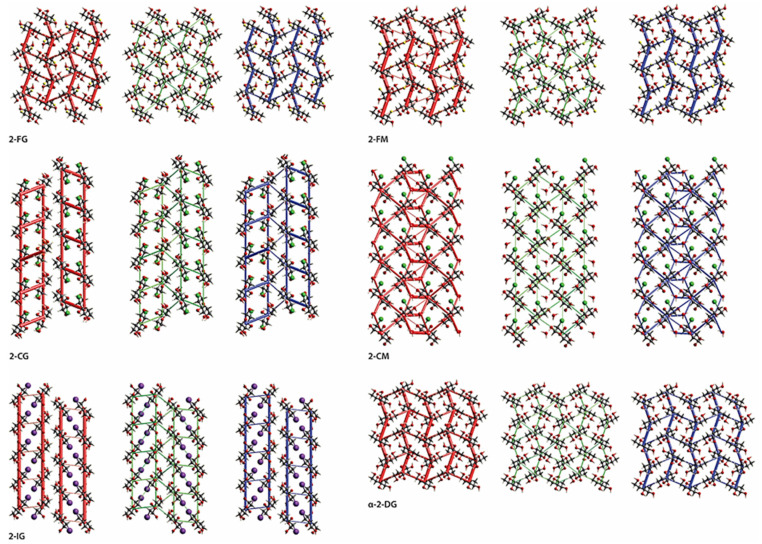
Energy frameworks viewed along *a* direction for the electrostatic (red) and the dispersion (green) components and the total interaction energy (blue).

**Figure 7 ijms-22-03720-f007:**
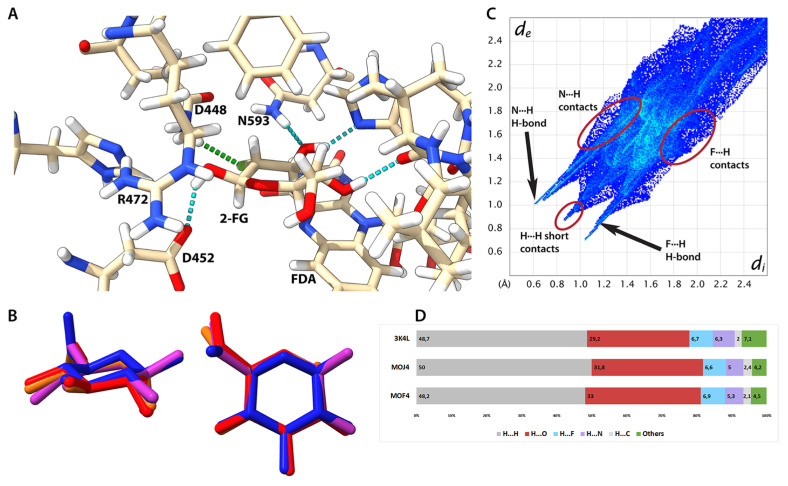
Interactions between 2-FG and selected mutants of P2Ox. (**A**) Structure of catalytic centre of 3K4L structure (some residues have been removed for the sake of ligand visibility). Canonical H-bonds are depicted as dotted cyan lines and F-H bond is depicted as a green dotted line. Residues having a substantial contribution (both negative and positive) to binding are labelled. (**B**) Superimposed structures of 2-FG from 3K4L (magenta), MOJ4 (red), and MOF4 (orange) compared to the structure from the crystalline state (blue). (**C**) Fingerprint plot for the complexed 2-FG in 3K4L structure. (**D**) Contribution of different X···X contacts for 2-FG protein complexes derived from their Hirshfeld surfaces.

**Table 1 ijms-22-03720-t001:** Cohesive energies for 2-DG and its derivatives.

Compound	Total Energy (kJ mol^−1^)
*α*-2-DG	−197.10
*β*-2-DG	−215.00
*β*-2-FG	−217.56
*β*-2-FM	−240.25
*β*-2-CG	−208.16
*β*-2-CM	−295.89
*α*-2-IG	−199.70

**Table 2 ijms-22-03720-t002:** Electrostatic interaction energies (kJ mol^−1^) between 2-FG and residues in the binding pocket for chosen ligand-protein complexes. A type of point mutation(s) is provided in parentheses.

Residue Type	Number	Interaction Energy [kJ mol^−1^]		
		3K4L (F454N)	4MOF (H450G)	4MOJ (H450G/V456C)
T	169	−56.0	−144.3	−140.3
A	171	−22.5	−22.3	−24.9
L	361	−26.7	−27.9	−30.8
Q	448	−73.4	−47.6	−55.9
H ^a^	450	+11.5	−3.4	−3.2
D	452	+419.1	+431.7	+494.5
R	472	+449.2	−440.6	−497.2
F	474	−53.1	+86.0	−59.4
L	545	+48.7	+34.0	+53.0
V ^b^	546	−19.3	−38.7	−32.7
H	548	−35.3	−5.6	−24.1
N	593	−160.3	−131.2	−155.3
FDA ^c^	801	−103.5	−188.9	−163.7
SUM		−512.3	−490.0	−635.1

^a^ replaced by G in 4MOF and 4MOJ. ^b^ replaced by C in 4MOJ. ^c^ dihydroflavine-adenine dinucleotide.

**Table 3 ijms-22-03720-t003:** Kinetic parameters (with D-glucose as electron donor and O_2_ as electron acceptor) of P2Ox mutants compared to the total interaction energy between protein and 2-FG.

Parameter ^a^	3K4L	4MOF	4MOJ
Total *E*_el_ [kJ mol^−1^]	−512.3	−490.0	−635.1
Michaelis constant (*K*_m_) [mM]	1.5 ± 0.1	0.939 ± 0.04	2.43 ± 0.27
Turnover number (*k*_cat_) [s^−1^]	12.0 ± 0.0	12.5 ± 0.2	16.8 ± 0.27
Specificity constant (*K*_m_/*k*_cat_) [mM s^−1^]	8.2	12.7	13.5

^a^ parameters were taken from the literature [44,45].

**Table 4 ijms-22-03720-t004:** Experimental parameters for data acquisition for the NMR experiments.

Parameter	COSY	^13^C-HSQC	HMBC	NOESY	^1^H	^13^C-HSQC-AP
Relaxation delay [s]	1	1.5	1	6	1	1.5
Number of transients	4	2	4	4	8	8
Acquisition time [s]	0.15	0.15	0.15	0.15	1.7	1.2
Number of increments	200	300	200	200	n.a.	400

## Data Availability

Not applicable.

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
