# Peer review of "Experimental and Computational Studies on Structure and Energetic Properties of Halogen Derivatives of 2-Deoxy-D-Glucose"

_ijms, 2021, doi:10.3390/ijms22073720_

Round 1
Reviewer 1 Report
In the present manuscript Pajak, Wozniak, Priebe and Ziemniak described very interesting experimental and computational studies on the structure and energetic properties of halogen derivatives of 2-deoxy-D-glucose, an important inhibitor of glycolysis, with anticancer action.
The manuscript is well written, clear and well-organized.
Only some minor typos were found and should be corrected before publication:
Some exemples are listed below:
Line 68, “dependend deseases”, should be replaced by “dependent diseases”
Line 72, “only a few their crystal structures”, should be replaced by “ only a few crystal structures”.
Lines 292, 293, “Both 2-FG and 2-FM forms…”, should be replaced by “Both 2-FG and 2-FM form…”
Line 306, please confirm if it is Gln or Glc
Line 728, replace “…in the the crystal…”, by “…in the crystal…”.
Line 742, replace “…in the biding cavity…” by “…in the binding cavity…”, and “…in its the crystal structure…” by “…in its crystal structure…”.
Line 483, please define UBDB.
The references are appropriate. Some need to be corrected according to the reference style in use;
Some examples:
Ref. 26, the journal title should appear after the manuscript title. The same corrections should be done for references 48 and 49.
Ref. 38, the journal name is J. Am. Chem Soc.
Ref.45, write the abbreviated journal name (J. Biol. Chem.).
In the title of the manuscripts cited in references 17, 30, 33, replace the unknown symbol by the correct letter.
Author Response
We acknowledge the reviewers' comments, who very accurately indicated the weak points of our paper. All comments were taken into account, and the changes visibly enriched the content of the paper. We hope that the reviewers will be satisfied with the current form of the manuscript.
Review 1
In the present manuscript Pajak, Wozniak, Priebe and Ziemniak described very interesting experimental and computational studies on the structure and energetic properties of halogen derivatives of 2-deoxy-D-glucose, an important inhibitor of glycolysis, with anticancer action. The manuscript is well written, clear and well-organized. Only some minor typos were found and should be corrected before publication:
Some examples are listed below:
- Line 68, “dependend deseases”, should be replaced by “dependent diseases”
- The line has been corrected.
- Line 72, “only a few their crystal structures”, should be replaced by “ only a few crystal structures”.
- The line has been corrected.
- Lines 292, 293, “Both 2-FG and 2-FM forms…”, should be replaced by “Both 2-FG and 2-FM form…”
- Lines have been corrected.
- Line 306, please confirm if it is Gln or Glc
- It should be Glc (glucose). The line has been corrected.
- Line 728, replace “…in the the crystal…”, by “…in the crystal…”.
- The line has been corrected.
- Line 742, replace “…in the biding cavity…” by “…in the binding cavity…”, and “…in its the crystal structure…” by “…in its crystal structure…”.
- The line has been corrected.
- Line 483, please define UBDB.
- UBDB abbreviation has been clarified.
- The references are appropriate. Some need to be corrected according to the reference style in use;Some examples:
Ref. 26, the journal title should appear after the manuscript title. The same corrections should be done for references 48 and 49.
Ref. 38, the journal name is J. Am. Chem Soc.
Ref.45, write the abbreviated journal name (J. Biol. Chem.).
-References have been corrected.
- In the title of the manuscripts cited in references 17, 30, 33, replace the unknown symbol by the correct letter.
- The unknown symbol was created during the conversion of the file to pdf format and replaced the alpha (a) symbol visible in the doc. version of the manuscript.
Reviewer 2 Report
The work is well-presented and rich of details for the area of interest. The spontaneous question is: What about the bromine derivatives? Is there any specific reason (synthesis, data already published or whatsoever) for their exclusion? They should be at least mentioned in the main text. Beyond that, I've found the reference #12 inappropriate as glucose (or glucose derivatives) is not the topic that article. Some other minor issue: Line 58, "relays" ... I guess you meant "relies".
Author Response
We acknowledge the reviewers' comments, who very accurately indicated the weak points of our paper. All comments were taken into account, and the changes visibly enriched the content of the paper. We hope that the reviewers will be satisfied with the current form of the manuscript.
Review 2
The work is well-presented and rich of details for the area of interest.
- The spontaneous question is: What about the bromine derivatives? Is there any specific reason (synthesis, data already published or whatsoever) for their exclusion? They should be at least mentioned in the main text.
- Due to the problems with optimizing the conditions and obtaining the appropriate crystals, 2-deoxy-2-iodo-D-mannppyranose (2-IM), 2-deoxy-2-bromo-D-glucopyranose (2-BG) and 2-deoxy-2-bromo-D-mannopyranose (2-BM) have not been analyzed so far. Thus, 2-IM and bromo-derivatives are still under scrutiny in our lab, and the results will be published in a separate paper.
The explanation was also introduced to the text in the introduction section.
- Beyond that, I've found the reference #12 inappropriate as glucose (or glucose derivatives) is not the topic that article.
- The citation was corrected as it was incorrectly cited in the wrong position. We believe, however, that cited article has its justification as 2-DG derivatives correspond in the research methodology to glucose derivatives.
- Some other minor issue: Line 58, "relays" ... I guess you meant "relies".
- The line has been corrected.